# Cell-Based Therapies for Solid Tumors: Challenges and Advances

**DOI:** 10.3390/ijms26125524

**Published:** 2025-06-09

**Authors:** Anna Smolarska, Zuzanna Kokoszka, Marcelina Naliwajko, Julia Strupczewska, Jędrzej Tondera, Maja Wiater, Roksana Orzechowska

**Affiliations:** 1Center of Cellular Immunotherapies, Warsaw University of Life Sciences, 02-787 Warsaw, Poland; 2Faculty of Biology and Biotechnology, Warsaw University of Life Sciences, 02-787 Warsaw, Poland

**Keywords:** cancer, solid tumors, cell-based therapy

## Abstract

Solid tumors pose significant therapeutic challenges due to their resistance to conventional treatments and the complexity of the tumor microenvironment. Cell-based immunotherapies offer a promising approach, enabling precise, personalized treatment through immune system modulation. This review explores several emerging cellular therapies for solid tumors, including tumor-infiltrating lymphocytes, T cell receptor-engineered T cells, CAR T cells, CAR natural killer cells, and macrophages. Tumor-infiltrating lymphocytes and their modified versions, T cell receptor-engineered T cells and CAR T cells, provide personalized immune responses, although their effectiveness can be limited by factors like variation in tumor antigens and the suppressive nature of the tumor environment. Natural killer cells engineered with chimeric receptors offer safer, non-major histocompatibility complex-restricted targeting, while modified macrophages exploit their natural ability to enter tumors and reshape the immune landscape. CAR-modified macrophages and macrophages conjugated with drugs are also considered as therapy for solid tumors. The review also examines the implications of autologous versus allogeneic cell sources. Autologous therapies ensure immunologic compatibility but are limited by scalability and manufacturing constraints. Allogeneic approaches offer “off-the-shelf” potential but require gene editing to avoid immune rejection. Integrating synthetic biology, gene editing, and combinatorial strategies will be essential to enhance efficacy and expand the clinical utility of cellular immunotherapies for solid tumors.

## 1. Introduction

Cancer remains one of the most rapidly progressing global health challenges. According to the World Health Organization (WHO), approximately 20 million new cancer cases and 9.7 million cancer-related deaths were reported in 2022 [1]. It is projected that by the year 2050, the global incidence of cancer will exceed 35 million new cases, representing a 77% increase compared to the estimated 20 million cases in 2022 [1]. These statistics underscore the critical need for continuously advancing novel, highly effective therapeutic strategies and the rigorous implementation of ongoing clinical trials. The sustained development of scientific research is essential for enhancing cancer treatment efficacy and transitioning from conventional therapies to more personalized approaches.

According to the National Cancer Institute’s Dictionary of Cancer Terms, solid tumors represent abnormal tissue proliferation, typically lacking cystic structures or fluid-filled areas. The management of solid tumors poses considerable challenges, particularly in terms of surgical resection, as well as resistance to conventional radiotherapy and chemotherapy, often leading to recurrence [2,3]. This has been linked to the presence of cancer stem-like cells (CSCs), a subpopulation of tumor cells with self-renewal capacity and the ability to differentiate into non-stem cancer cells (NSCCs), which constitute the bulk of the tumor’s mass [4,5,6]. A comprehensive understanding of CSC biology is crucial for developing therapeutic strategies targeting these resilient cancer cell populations. A deeper understanding and advancement of cell-based therapies can potentially enhance solid tumor treatment by specifically targeting both CSCs and NSCCs, thereby improving tumor eradication and reducing the risk of recurrence.

A key advantage of cell-based therapies is their ability to provide personalized treatment modalities tailored to individual patients [7]. Unlike conventional first-line cancer treatments, which often exhibit variable efficacy across different patients, cell therapies are designed to enhance patient-specific immune responses [8]. The primary objective of cell therapy for solid tumors is the precise identification and targeting of malignant cells, coupled with the activation, stimulation, and support of the immune system to eliminate tumor cells effectively [9].

Cell-based therapies encompass a diverse range of strategies, including both stem and non-stem cell approaches and single-cell and multicellular formulations with distinct immunophenotypic characteristics and mechanisms of action [10]. The immune system plays a pivotal role in recognizing and eradicating malignancies; however, cancer cells have evolved sophisticated mechanisms to evade immune surveillance [11]. Moreover, malignant cells can exploit immune system components to promote tumor progression and immune escape [12]. Cell therapies aim to augment or reprogram immune system components to counteract tumor-induced immunosuppression and enhance anti-tumor efficacy [13,14,15]. Significant progress has been made in elucidating mechanisms of cancer immune evasion and tumor–host interactions [16,17,18,19,20]. Still, further advancements are required to develop more effective strategies to counteract the ability of cancer cells to evade immune surveillance and resist immune-mediated elimination.

According to a report by Precedence Research, the global cell therapy market was valued at approximately USD 6.04 billion in 2024 and is projected to expand at a compound annual growth rate (CAGR) of 22.96% through 2034, reaching USD 47.72 billion. The oncology segment accounted for approximately 29.8% of the global cell therapy market share in 2023. This suggests that cell-based cancer therapies were valued at approximately USD 1.8 billion that year. Despite the promising potential of cell therapies, the majority remain in early-phase clinical trials (phases I and II), primarily due to safety concerns and adverse effects that hinder their integration into standard oncology treatment regimens [10,21]. Additionally, the highly individualized nature of these therapies contributes to their substantial costs, posing further challenges to widespread clinical adoption [22].

This review will explore cellular therapies with potential applications in solid tumor treatment, including tumor-infiltrating lymphocytes (TILs) and T cell receptor-engineered T cells. Furthermore, innovative therapies utilizing chimeric antigen receptor (CAR) T cells, CAR natural killer cells (CAR NK), and macrophages will be discussed. Their mechanisms of action are shown in the figure (Figure 1). The discussion will also encompass the implications of autologous versus allogeneic therapeutic approaches in the context of cellular immunotherapy for solid tumors.

## 2. Cell-Based Therapies in Solid Tumor Treatment

Innovative approaches in cancer treatment have led to the development of advanced therapeutic strategies, with immunotherapy standing out as a promising modality. Immunotherapy often involves T lymphocytes, which are crucial immune cells responsible for coordinating immune responses and eliminating pathogen-infected and cancerous cells [23,24]. Cell-based therapies for solid tumors increasingly incorporate other immune effector cells, such as NK cells and macrophages. These alternative cell types offer complementary mechanisms of action, such as innate cytotoxicity, phagocytosis, and tumor microenvironment remodeling, which may overcome resistance pathways and physical barriers that limit T cell efficacy [25,26]. The most important milestones in the development of the described cell-based therapeutic modalities—TILs, TCR T, CAR T, CAR NK, and modified macrophages—are summarized in Figure 2. This timeline highlights key translational breakthroughs, from first-in-human applications to pivotal regulatory approvals and ongoing clinical advances.

### 2.1. Tumor-Infiltrating Lymphocytes

Among the most promising strategies of novel therapeutic approaches in solid tumor treatment is using TILs—a component of the adaptive immune system—as a form of personalized immunotherapy [32]. TILs can be isolated directly from a patient’s tumor, expanded ex vivo to clinically significant numbers, and subsequently reinfused into the patient, typically following lymphodepleting chemotherapy [33]. This approach aims to potentiate the endogenous anti-tumor immune response and overcome immunosuppressive barriers, including dense stromal architecture, hypoxia, inhibitory cytokines (e.g., TGF-β, IL-10), and immune checkpoint expression (e.g., PD-L1, TIM-3, LAG-3) within the tumor microenvironment (TME).

#### 2.1.1. Mechanism

The standard approach to TIL-based therapy begins with surgical resection of the tumor. TILs are isolated and cultured ex vivo in the presence of high concentrations of interleukin-2 (IL-2) to promote their activation and proliferation. After sufficient expansion, these TILs are infused back into the patient, typically following a lymphodepleting regimen to eliminate endogenous immune cells to enhance TIL engraftment, persistence, and functionality [33].

TILs primarily comprise T lymphocytes, especially CD8^+^ cytotoxic T lymphocytes (CTLs) and CD4^+^ helper T cells, although other immune subsets, including B cells and natural killer (NK) cells, may also be present [34,35]. CTLs play a central role in recognizing and destroying malignant cells through interactions with tumor-specific antigens presented by major histocompatibility complex (MHC) molecules. TILs are particularly valuable due to their intrinsic specificity for tumor-associated antigens, including neoantigens derived from tumor-specific mutations, enabling precise and potent immune targeting of cancer cells [36,37].

The TME, comprising malignant cells, stromal elements, immune infiltrates, and extracellular matrix components, presents numerous challenges to effective TIL function. The immunosuppressive nature of the TME is mediated by elements like regulatory T cells (Tregs), myeloid-derived suppressor cells (MDSCs), and immunomodulatory cytokines, like transforming growth factor-beta (TGF-β) and interleukin-10 (IL-10). It can significantly impair the effector functions of TILs [38,39,40]. Despite these barriers, the presence of TILs within tumors has been positively correlated with improved clinical outcomes in various malignancies, including melanoma, colorectal cancer, and ovarian cancer [41,42,43].

#### 2.1.2. Clinical Data

Clinical trials, particularly in metastatic melanoma, have demonstrated that TIL therapy can induce durable clinical responses, with some patients achieving complete remission [44,45]. TILs can overcome the TME’s immunosuppressive barriers by reintroducing a large pool of tumor-reactive T cells with restored effector function and enhanced cytotoxicity. Upon reinfusion, TILs possess the capacity to penetrate immunologically “cold” tumors, resist local T cell dysfunction, and exert targeted cytolytic activity even in the presence of immunosuppressive mediators. Furthermore, the polyclonality of TIL populations allows for recognition of multiple tumor-associated neoantigens, reducing the likelihood of immune escape and broadening anti-tumor coverage. Clinical studies have shown that TIL therapy can effectively remodel the TME, shifting it toward a pro-inflammatory and immune-permissive state, thereby facilitating durable responses in checkpoint-refractory solid tumors [46,47,48]. An example of these abilities to overcome barriers is the FDA-approved product Lifileucel (Amtagvi^TM^) for PD-1-refractory metastatic melanoma, which restores anti-tumor immunity through ex vivo-expanded autologous lymphocytes capable of mediating durable responses despite prior checkpoint inhibitor failure [49].

These promising outcomes have stimulated efforts to adapt TIL therapy to other solid tumor types. However, several challenges persist, including patient selection criteria, variability in TIL expansion success, and potential treatment-related toxicities [50,51].

#### 2.1.3. Limitations

Despite their clinical promise, TILs face several limitations that constrain their broad applicability. First, TIL therapy requires extensive ex vivo expansion from surgically resected tumor tissue, a process that is both labor-intensive and not always feasible for patients with inaccessible or low-yield tumors [51]. Additionally, the composition and functionality of TILs can vary widely among patients, often influenced by the immunosuppressive TME, which may render them exhausted or antigen-non-specific [52,53]. Moreover, TILs are largely HLA-restricted, limiting their effectiveness across genetically diverse patient populations, and their persistence after infusion is often limited without adjunctive IL-2 support, which can itself cause severe toxicities [54].

To address these limitations, researchers are investigating next-generation strategies, such as genetic modification of TILs to enhance persistence and cytotoxicity, integration of an immune checkpoint blockade to mitigate TME-induced inhibition, and combinatorial regimens incorporating TILs with targeted therapies, vaccines, or cytokine support. These innovations aim to improve the efficacy and broaden the applicability of TIL-based immunotherapies [55,56].

TIL therapy represents a compelling and evolving approach to treating solid tumors. With their natural ability to recognize and eliminate cancer cells and ongoing technological and therapeutic advancements, TILs offer significant promise, particularly for patients with treatment-refractory malignancies.

### 2.2. T Cell Receptor-Engineered T Cells (TCR T)

T cell receptor-engineered T (TCR T) cells are genetically modified autologous or allogeneic T lymphocytes that express tumor-specific T cell receptors capable of recognizing intracellular antigens presented by MHC molecules. Their anti-tumor function is mediated through the specific recognition of tumor-associated antigens (TAAs) presented by MHC-I molecules on the surface of malignant cells, followed by targeted cytolysis [57]. This enables their application in tumors like melanoma, synovial sarcoma, and non-small cell lung cancer (NSCLC), where shared or patient-specific HLA-restricted tumor epitopes (e.g., NY-ESO-1, MAGE-A4, KRAS-G12D) are present [58,59,60].

Adaptive T cell therapy employing ex vivo expanded TRCs represents one of the earliest clinical applications of cell-based immunotherapy. Initial successes were observed in virus-associated malignancies, including Epstein–Barr virus (EBV)-positive lymphomas and nasopharyngeal carcinoma, where the immunodominance of viral antigens facilitates robust T cell targeting [61]. In these protocols, autologous CTLs are typically generated through in vitro stimulation of peripheral blood mononuclear cells (PBMCs) with antigen-loaded dendritic cells or irradiated tumor cells, followed by cytokine-supported expansion and subsequent reinfusion into the patient [61].

#### 2.2.1. Mechanism

TCR T cells are generated by isolating peripheral blood T lymphocytes from a patient or donor and genetically modifying them to express a tumor-specific TCR with high affinity for antigens presented by MHC molecules on tumor cells. The TCR transgene is typically introduced using viral vectors, such as lentivirus or retrovirus. After transduction, the modified T cells are expanded ex vivo and infused back into the patient following a lymphodepleting conditioning regimen to enhance their persistence and function [62]. The engineering of T cells to express TCRs has enabled precise targeting and elimination of malignant cells [63].

#### 2.2.2. Clinical Data

This approach has led to durable clinical responses in hematological malignancies and is increasingly being adapted for the treatment of solid tumors [63]. TCR T cells have shown promise in clinical trials, particularly in solid tumors where CAR T cell therapy has had limited success. Trials using NY-ESO-1-specific TCR T cells in synovial sarcoma and melanoma have demonstrated durable responses in some patients [64,65]. However, their effectiveness is influenced by factors like HLA restriction, tumor antigen heterogeneity, and immune evasion via MHC downregulation.

#### 2.2.3. Limitations

Despite these advances, the therapeutic efficacy of TRCs, like other CTLs, in solid tumors remains constrained by multiple immunologic and tumor-intrinsic barriers, including antigenic heterogeneity, defective antigen presentation machinery, and the immunosuppressive TME, which collectively impair TCR T cells’ infiltration, survival, and function [66]. To circumvent these obstacles, recent strategies have focused on the genetic engineering of TRCs to augment their persistence and resistance to immunosuppressive signals, such as through dominant-negative TGF-β receptor expression or the incorporation of costimulatory signaling domains to enhance effector function and proliferation. CAR T cells are an example of another approach for T cell application in cancer treatment [67,68,69].

### 2.3. Chimeric Antigen Receptor T Cells (CAR T)

The effectiveness of chimeric antigen receptor CAR T cell therapy lies in the genetic modification of T cells to express CARs, enabling them to recognize and target specific antigens present on the surface of tumor cells, thereby activating T cells independently of the MHC [70,71]. This enhancement of T cell function presents new treatment avenues for patients with certain cancers.

#### 2.3.1. Mechanism

CARs typically consist of three key components: an extracellular antigen-binding domain, a transmembrane domain, and an intracellular signaling domain [72,73]. The extracellular domain, responsible for antigen recognition, imparts specificity to CAR T cells, typically derived from single-chain variable fragments (scFvs) of antibodies that target TAAs or tumor-specific antigens (TSAs), such as HER2, mesothelin, EGFRvIII, GD2, and Claudin 18.2. The transmembrane domain anchors the CAR to the T cell membrane, ensuring its proper stability and signaling functionality [74]. The intracellular signaling domain transmits activation signals following antigen recognition and typically includes the CD3ζ domain, which can be combined with domains like CD28, 4-1BB (CD137), or OX40 (CD134) [75,76]. Including these co-stimulatory domains improves CAR T cell persistence, proliferation, and anti-tumor efficacy.

CAR T cells have evolved through several generations, with first-generation CARs incorporating only the CD3ζ domain, which has shown reduced in vivo efficacy [77]. Second-generation CARs incorporate a single co-stimulatory domain [78], while third-generation CARs contain two or more co-stimulatory domains [79]. Fourth-generation CARs, also known as TRUCKs (T cells Redirected for Universal Cytokine-mediated Killing), are designed to release cytokines upon antigen recognition, thus enhancing their tumor-killing potential [80,81].

#### 2.3.2. Clinical Data

CAR T cell therapy has shown remarkable efficacy in treating hematologic malignancies, such as leukemia and lymphoma, where cancer cells circulate in the blood or the lymphatic system [80]. However, the treatment landscape becomes more complex when addressing solid tumors. Unlike hematologic cancers, solid tumors present unique challenges hindering CAR T cell therapy. These include immune evasion mechanisms, which allow tumors to avoid detection by the immune system, and a hostile TME that can inhibit CAR T cell function [14,68]. The TME includes dense extracellular matrices (ECM), non-cancerous stromal cells, and immune suppressive factors that obstruct CAR T cell infiltration and survival. Additionally, the abnormal vasculature often found in solid tumors limits the effective delivery of CAR T cells [14].

Although CAR T cell therapy has shown remarkable success in hematologic malignancies, several early-phase clinical trials have also demonstrated encouraging efficacy in solid tumors. For example, GD2-targeted CAR T cells have led to objective responses in patients with neuroblastoma and diffuse midline gliomas, including a case of radiographic remission in a child with pontine glioma [82]. Claudin 18.2 CAR T cells have shown promise in gastric and pancreatic cancers, with high response rates reported in a phase I study [83]. Furthermore, mesothelin-directed CAR T cells have induced disease stabilization in mesothelioma and pancreatic cancer when combined with regional or intrapleural delivery approaches [84]. These studies illustrate the feasibility and potential of CAR T therapy in solid tumors when combined with tumor-specific targeting, optimized delivery, and resistance mitigation strategies.

#### 2.3.3. Limitations

One of the most significant hurdles in CAR T cell therapy for solid tumors is the identification of suitable target antigens that are both highly expressed on tumor cells and absent or minimally present in normal tissues [85,86,87]. Prominent TAAs under investigation include HER2, mesothelin, EGFRvIII, GD2, and Claudin 18.2, each associated with specific tumor types, such as breast, ovarian, glioblastoma, neuroblastoma, and gastric cancers, respectively [84,88,89,90,91]. Many tumor-associated proteins are also found in healthy tissues, raising the risk of off-target effects, including those tragically observed in early trials of HER2-CAR T cell related fatality due to cytokine-storm-like syndrome, accompanied by rapid respiratory failure and multi-organ dysfunction. Post-mortem analyses suggested that the CAR T cells had reacted not only with tumor cells but also with low-level HER2 expression in the pulmonary epithelium, leading to massive immune activation and widespread tissue damage [85]. Furthermore, solid tumors exhibit antigen heterogeneity, with varying levels or types of antigens across different tumor cells, making it challenging for CAR T cells to target and eliminate all malignant cells [92].

To address these challenges, various strategies have been explored. These include engineering CAR T cells to recognize and target multiple antigens [93], incorporating enzymes into CAR T cells that can degrade the ECM and facilitate better tumor infiltration [94], and identifying novel, highly specific antigens unique to cancer cells to improve targeting precision and reduce damage to healthy tissues [95]. Developing personalized CAR T cell therapies based on the unique antigen profile of a patient’s tumor is another promising strategy for overcoming antigen heterogeneity.

An emerging strategy to enhance the efficacy of CAR T cells in solid tumors involves co-targeting not only malignant cells but also the tumor stroma, particularly cancer-associated fibroblasts (CAFs), which play a central role in immune exclusion and therapeutic resistance. Huang et al. (2025) reported the development of a universal CAR T cell platform capable of redirecting cytotoxic activity against both tumor cells and CAFs through the use of bispecific adapter molecules [96]. Their design enabled the simultaneous engagement of tumor-associated antigens and fibroblast activation protein, a surface marker enriched on CAFs, resulting in synergistic tumor killing and stromal disruption in preclinical models of solid cancer. Importantly, the universal targeting system allowed for dynamic modulation of specificity and intensity via adapter dosing, offering a tunable approach to reduce off-tumor effects. This approach aligns with broader developments in modular CAR architectures, such as those described in the WO2024191887A2 patent, which detail adapter-mediated systems engineered to engage multiple components of the TME concurrently, thereby addressing spatial heterogeneity and immune evasion mechanisms inherent to solid tumors.

Notably, bispecific constructs targeting disialoganglioside (GD2) and prostate-specific membrane antigen (PSMA) have gained attention for their application in neuroectodermal and prostate malignancies, respectively. Preclinical models and early-phase clinical trials have shown that co-targeting GD2 and PSMA enhances T cell infiltration and cytotoxicity while potentially limiting immune escape mechanisms associated with monovalent CAR therapies. Ongoing trials, such as NCT05437315, continue to evaluate the safety, persistence, and anti-tumor activity of GD2/PSMA bispecific CAR T cells in solid tumors. These findings highlight the clinical feasibility of dual-targeting approaches and support the broader development of bispecific CAR platforms to overcome tumor antigen heterogeneity.

Ongoing investigations into CAR T cell therapy for solid tumors are being conducted across various cancers, including glioblastoma (NCT05577091, NCT04077866, NCT05353530), renal (NCT05420519, NCT04969354), prostate (NCT03873805), ovarian (NCT06305299, NCT05211557), and lung cancers (NCT06903117). Although early clinical trials have not yielded the same level of success as seen in hematologic malignancies [97,98], advances in CAR engineering and a deeper understanding of the challenges posed by the TME will drive progress in this field.

### 2.4. CAR Natural Killer Cells

CAR NK cells represent an emerging and innovative modality in the field of adoptive cell therapy designed to augment the innate cytotoxic potential of NK cells through tumor-specific redirection. In contrast to CAR T cells, which have demonstrated robust efficacy in hematologic malignancies, CAR NK cells offer several distinct immunological and safety advantages, including MHC-independent tumor recognition, a negligible risk of graft-versus-host disease (GvHD), and a lower incidence of cytokine release syndrome (CRS) and neurotoxicity [31,99].

#### 2.4.1. Mechanism

CAR NK cells can be generated from various sources, including peripheral blood, umbilical cord blood, induced pluripotent stem cells (iPSCs), and established NK cell lines, such as NK-92. Each source offers distinct advantages regarding cytotoxic efficacy, expandability, and suitability for genetic modification [100]. For example, NK-92 cells are readily expandable and amenable to CAR transduction but require irradiation prior to infusion to prevent in vivo proliferation due to their malignant origin [101].

NK cells exert innate anti-tumor activity via a balance of activating and inhibitory receptor signaling, culminating in the direct lysis of transformed cells through perforin- and granzyme-mediated apoptosis. Additionally, NK cells mediate antibody-dependent cellular cytotoxicity (ADCC) through the expression of the FcγRIIIa receptor (CD16), further expanding their cytotoxic repertoire [102]. Incorporating CAR constructs into NK cells significantly enhances their tumor specificity and cytolytic potency, allowing antigen-directed killing via synthetic antibody-like scFv domains while preserving their endogenous effector mechanisms [31].

#### 2.4.2. Clinical Data

CAR-NK cells’ clinical feasibility and therapeutic potential were first demonstrated in a landmark phase I/II clinical trial involving patients with relapsed or refractory CD19-positive lymphoid malignancies. In that study, umbilical-cord-blood-derived CAR NK cells, co-expressing interleukin-15 (IL-15) and an inducible caspase-9 safety switch, yielded objective responses in 8 out of 11 patients (73%), with no incidence of CRS, neurotoxicity, or GvHD, highlighting the favorable safety profile of this therapeutic platform [31].

Still, the application of CAR NK cells in solid tumors has been met with challenges analogous to those encountered in CAR T cell therapy. These include antigenic heterogeneity, physical and stromal barriers to infiltration, and immunosuppressive elements within the TME. Nonetheless, preclinical models have demonstrated encouraging anti-tumor activity of CAR NK cells targeting HER2, mesothelin, and epidermal growth factor receptor (EGFR) in malignancies like glioblastoma, breast cancer, and ovarian cancer [103,104,105,106].

#### 2.4.3. Limitations

To enhance efficacy in the context of solid tumors, current efforts are focused on the development of “armored” CAR NK cells equipped with transgenes encoding pro-survival cytokines (e.g., IL-15), dominant-negative TGF-β receptors, and chemokine receptors to improve trafficking and persistence in the TME [107]. Moreover, generating “off-the-shelf” CAR-NK cell products from allogeneic sources, such as cord blood or iPSCs, presents a scalable and potentially more accessible therapeutic alternative to autologous CAR T cell therapy [108].

### 2.5. Macrophages

In response to challenges in applying CAR T cells in solid tumors [109,110,111], macrophages have emerged as a promising alternative effector cell type for CAR engineering owing to their intrinsic tumor-homing capacity and versatile immunomodulatory functions [112,113].

#### 2.5.1. Mechanism

Recent advances have enabled the genetic modification of human macrophages to express CARs, thereby redirecting their phagocytic activity toward tumor cells [114]. Notably, using a chimeric adenoviral vector has facilitated efficient gene delivery by circumventing innate immune barriers typically encountered in macrophage transduction. This approach has successfully reprogrammed macrophages to adopt a sustained pro-inflammatory (M1-like) phenotype, enhancing their anti-tumor potential [114].

#### 2.5.2. Clinical Data

In vitro studies have demonstrated that CAR macrophages (CAR M) exhibit antigen-specific phagocytosis and cytotoxicity against tumor cells [26]. Moreover, CAR M therapy promotes a shift in the macrophage phenotype from immunosuppressive M2 to pro-inflammatory M1, elevates pro-inflammatory cytokine and chemokine production, enhances antigen presentation, and recruits T cells into the tumor microenvironment. These effects collectively contribute to remodeling of the tumor milieu and potentiate adaptive anti-tumor immunity. In in vivo studies using humanized mouse models, CAR Ms have been shown to induce robust anti-tumor responses and promote T-lymphocyte infiltration and activity, further supporting their therapeutic potential in solid tumors [115].

Currently, several biotechnology companies are advancing CAR M platforms with distinct technological strategies:

SIRPαnt Immunotherapeutics has developed SIRPant-M^TM^ (SI-101), an autologous, non-genetically engineered macrophage therapy utilizing the PhagoAct^TM^ platform. This method activates and educates patient-derived macrophages ex vivo to recognize and eliminate cancer cells through intrinsic immune mechanisms. SIRPant-M^TM^ engages both cellular and humoral immune responses and promotes long-term anti-tumor immunity. The therapy is presently in preclinical development in relapsed or refractory non-Hodgkin lymphoma (ClinicalTrials.gov ID NCT05967416). SIRPant Immunotherapeutics announced in late 2023 that it had received FDA clearance to begin a phase 1 clinical trial of SIRPant-M^TM^ for the treatment of various solid tumors, such as head and neck cancer, non-melanoma skin cancers, bladder and kidney cancers, low-grade prostate cancer, triple-negative breast cancer, and certain sarcomas. Under the newly cleared Investigational New Drug, the company plans to initiate clinical investigation of SIRPant-M as a monotherapy and in combination with other immuno-stimulatory modalities, such as radiotherapy and immune checkpoint inhibitors, for the treatment of select solid tumor indications.

Carisma Therapeutics is developing CT-0508, a genetically engineered CAR M product targeting HER2-positive solid tumors (ClinicalTrials.gov ID NCT04660929). Preclinical data suggest that CT-0508 can effectively infiltrate tumors, kill malignant cells, reprogram the tumor microenvironment, and facilitate the recruitment of adaptive immune cells. The study demonstrated a favorable safety profile with no dose-limiting toxicities. Approximately 44% of patients with HER2 3+ tumors achieved stable disease as the best overall response. However, the development of this platform has paused as the company shifts its focus to other technologies.

Cellis has introduced a novel platform called Macrophage Drug Conjugates (MDCs). This approach leverages macrophages loaded with ferritin-based drug complexes for targeted delivery into tumor cells [116]. Central to this platform is the TRAIN (Targeted Reprogramming of Antigen-presenting Immune Networks) mechanism, which enables precise cytoplasmic delivery of therapeutic payloads, inducing tumor cell death and immunogenic modulation of the microenvironment. MDCs have demonstrated enhanced survival, reduced tumor burden, and decreased metastases in preclinical models of solid tumors. The company is preparing to initiate first-in-human phase 1 clinical trials for its lead candidate, MDC-735, in the second quarter of 2025. These trials are planned to take place in Switzerland, Germany, and Poland, targeting multiple solid tumors, including glioblastoma, ovarian, bladder, lung, and head and neck cancers. The technology is on track to enter clinical trials in Q2 2025.

Collectively, these innovations underscore the potential of CAR macrophage therapies as next-generation cell-based immunotherapies, particularly in the context of solid tumors, where conventional CAR T strategies have faced limitations.

## 3. Current Clinical Trials of Cell-Based Therapies Against Solid Tumors

Clinical trials investigating cell-based therapies in solid tumors are essential for advancing the field of cancer immunotherapy beyond hematologic malignancies. These trials provide critical insights into the safety, feasibility, and preliminary efficacy of novel therapeutic modalities, such as TILs, TCR T, CAR T cells, CAR NK cells, and engineered macrophages, in the context of the immunosuppressive TME. As solid tumors account for the vast majority of cancer-related morbidity and mortality, the successful translation of cell-based approaches could significantly improve clinical outcomes. Clinical trials using cell-based therapies in solid tumor therapy conducted to date are summarized in Table 1.

## 4. Advances and Challenges in Cell-Based Therapy for Solid Tumors

Cell-based immunotherapies represent a promising approach for the treatment of various malignancies. Among these, CAR T therapy has shown potential efficacy against multiple cancer types, including pancreatic, colorectal, liver, lung, and gastric cancers. Despite this progress, significant challenges remain in the application of CAR T therapies to solid tumors. One major obstacle is the limited trafficking and infiltration of immune cells into solid tumor tissues. The TME exhibits strong immunosuppressive characteristics, severely impairing T cells’ anti-tumor activity [109].

Aberrant tumor vasculature further impedes T cell trafficking. Dysregulated angiogenesis, characterized by an imbalance between pro- and anti-angiogenic factors, forms large, unbranched vessels that fail to adequately perfuse the tumor tissue [117]. This poor perfusion results in hypoxia, increased levels of reactive oxygen species, and the activation of hypoxia-inducible factors (HIF1A and HIF2A), which promote the expression of immunosuppressive cytokines and growth factors [118].

Recent strategies to enhance T cell infiltration include tumor necrosis factor superfamily member 14 (LIGHT) expression on T cells. LIGHT expression induces the formation of tertiary lymphoid structures (TLSs) within tumors, significantly boosting local anti-tumor immune responses [119]. TLSs, which represent organized lymphoid aggregates within non-lymphoid tissues, facilitate enhanced immune surveillance and activation [120].

Moreover, when combined with vascular targeting peptides (VTPs), LIGHT expression contributes to vascular normalization within tumors. Studies in murine Lewis lung carcinoma and melanoma models have demonstrated that LIGHT-VTP treatment reduced hypoxia and improved tumor perfusion [121]. Similar findings have been reported in models of primary brain cancer [122], highlighting the potential of this approach to improve immune cell infiltration into solid tumors.

Another major obstacle is the role of immune checkpoints in the TME, which protect healthy tissues from immune-mediated damage but simultaneously limit anti-tumor responses [123]. Specifically, programmed cell death protein 1 (PD-1) expressed on activated T cells binds to its ligand PD-L1, which is often overexpressed by tumor-associated macrophages (TAMs) and other cells in the TME [124].

To counteract PD-1/PD-L1-mediated immunosuppression, small-molecule inhibitors, such as styryl carbamates, have been explored. These compounds, characterized by chemical stability and cellular permeability, can disrupt PD-1/PD-L1 interactions. In vitro co-cultures of cancer cells and T cells treated with styryl carbamates exhibited decreased cancer cell viability without significantly affecting immune cell survival. Moreover, styryl carbamates reduced PD-L1 expression and the formation of the PD-1/PD-L1 complex [125].

Additionally, CAR T cells engineered to secrete PD-1-blocking scFvs have demonstrated enhanced anti-tumor efficacy. In murine models of human lymphoma, treatment with such engineered CAR T cells resulted in increased tumor cell lysis and a reduction in PD-1 expression on T cell surfaces compared to control groups [126].

IL-15 co-expression in CAR-engineered immune cells has emerged as a strategy to enhance in vivo persistence, metabolic fitness, and anti-tumor function, particularly in NK and T cell platforms. Several preclinical and early-phase clinical studies have demonstrated that IL-15 promotes homeostatic proliferation and supports long-term survival of CAR T and CAR NK cells, translating into improved tumor control in solid and hematologic malignancies [31,127]. However, the inclusion of IL-15 is not without controversy. Elevated systemic levels of IL-15—especially when produced in an unregulated or constitutive manner—have been associated with increased risks of cytokine-mediated toxicities, including lymphoproliferation, neuroinflammation, and symptoms resembling CRS [128]. Furthermore, the effects of IL-15 appear to be highly context dependent, varying with the immune cell type, tumor microenvironment, route of administration, and vector design. Some studies report enhanced persistence without toxicity when IL-15 is membrane-bound or regulated by inducible systems, while others note adverse inflammatory consequences with secreted IL-15 constructs [129]. Thus, while IL-15 co-expression holds considerable therapeutic promise, its incorporation into CAR platforms requires precise control mechanisms and rigorous safety evaluation to balance efficacy with tolerability.

Together, these strategies provide promising avenues to overcome the hostile TME and enhance the therapeutic efficacy of CAR T cell therapies in the treatment of solid tumors.

## 5. The Implications of Autologous Versus Allogeneic Therapeutic Approaches in the Context of Cellular Immunotherapy for Solid Tumors

The selection of autologous versus allogeneic cellular sources constitutes a pivotal consideration in the development of adoptive immunotherapies for solid tumors. These two paradigms exhibit distinct immunological, logistical, and translational characteristics, each profoundly influencing therapeutic efficacy, manufacturing scalability, clinical feasibility, and patient accessibility.

Autologous cellular immunotherapy involves the procurement and ex vivo expansion or genetic modification of immune effector cells derived from the patient, followed by reinfusion of the modified product into the same individual. This approach underpins several advanced therapeutic strategies, including tumor-infiltrating lymphocyte (TIL) therapy and autologous CAR T cell therapy. A primary advantage of autologous approaches is their inherent immunological compatibility, thereby mitigating the risk of GvHD and recipient immune rejection. Moreover, the tumor specificity afforded by TILs—selected directly from the tumor microenvironment—permits the targeting of patient-specific neoantigens, offering a high degree of personalization particularly relevant for genetically complex solid tumors [130]. Nevertheless, autologous cell therapies are encumbered by substantial manufacturing constraints, including individualized processing workflows, prolonged production timelines, and variable product quality, which may be adversely affected by the patient’s prior treatments or immunocompromised status. These limitations pose significant barriers to scalability and timely clinical implementation.

Conversely, allogeneic cell therapies utilize immune cells sourced from healthy donors, which are expanded and genetically engineered for therapeutic administration to unrelated recipients [131,132]. This approach offers several practical advantages, chief among them the feasibility of developing standardized, “off-the-shelf” cell products that can be cryopreserved, stored, and deployed rapidly upon clinical need [133]. Allogeneic therapies significantly reduce per-patient manufacturing costs and facilitate batch processing, thus enabling broader dissemination and accessibility. In addition, donor-derived cells can be pre-selected for optimal functionality, cytotoxic potential, and fitness. However, the immunogenicity of allogeneic products presents a major obstacle. Unmodified donor T cells can induce GvHD, while host immune rejection may limit the persistence and efficacy of the therapeutic cells. To mitigate these risks, advanced gene-editing strategies, such as T cell receptor disruption, β2-microglobulin knockout, and HLA class I/II silencing, have been employed to reduce immunological incompatibility and extend in vivo durability [134]. Recent advancements in allogeneic NK cell therapy have been significantly driven by the development of robust ex vivo expansion platforms, particularly those employing genetically engineered feeder cell systems. The use of irradiated K562 cells modified to express membrane-bound interleukin-15 (mbIL-15) and 4-1BB ligand has markedly enhanced NK cell proliferation, survival, and cytotoxicity against tumor targets [135]. These feeder-based protocols enable the generation of clinical-scale NK cell populations with high purity, potent activation markers, and reduced exhaustion phenotypes, forming the backbone of several current good manufacturing practice-compliant protocols [31]. More recently, feeder cells have been further optimized to express mbIL-21, which supports greater NK cell expansion and maintains a less differentiated phenotype compared to IL-15-driven systems [136,137]. These innovations have laid the groundwork for scalable, off-the-shelf NK cell products that are being tested in early-phase clinical trials for both hematological and solid malignancies.

The application of autologous and allogeneic modalities to solid tumors is further complicated by tumor-intrinsic features impairing immune cell function and trafficking. These include pronounced antigenic heterogeneity, an immunosuppressive TME characterized by regulatory immune subsets and inhibitory cytokines, and a dense stromal architecture constituting a formidable physical barrier to cell infiltration [66]. While autologous therapies may offer superior tumor specificity and functional adaptability due to their endogenous origin, allogeneic approaches benefit from logistical flexibility and the potential for large-scale manufacturing. Notably, the development of allogeneic CAR NK cells and iPSC-derived immune effectors offers a promising avenue to bridge the gap between efficacy and accessibility by combining low immunogenicity with uniformity in therapeutic product composition. Allogeneic CAR NK cells have shown encouraging results in first-in-human trials, including high response rates (e.g., 73%) and minimal toxicity in hematologic malignancies [31]. Similarly, iPSC-derived NK cells have demonstrated consistent phenotype, potent cytotoxicity, and enhanced scalability in preclinical and early-phase clinical studies [138].

In conclusion, both autologous and allogeneic cell-based therapies possess unique and complementary strengths, and the optimal platform may vary depending on clinical context, tumor biology, therapeutic urgency, and infrastructure capabilities. The integration of next-generation technologies, including CRISPR-mediated gene editing, synthetic biology platforms, and combination immunotherapeutic regimens, will be essential to surmount existing barriers and maximize the therapeutic potential of cellular immunotherapy in solid tumors [139,140].

## 6. Conclusions

Cell-based immunotherapies have emerged as a rapidly advancing class of therapeutic strategies that could redefine the treatment paradigm for solid tumors. These approaches—including tumor-infiltrating lymphocytes, T cell receptor-engineered T cells, chimeric antigen receptor-engineered T cells, natural killer cells, and modified macrophages—offer the capacity for antigen-specific tumor targeting, immunological memory, and durable clinical responses. However, the translation of these therapies from hematologic malignancies to solid tumors remains impeded by a multitude of tumor-intrinsic and extrinsic barriers.

Key limitations include the heterogeneity of tumor antigen expression, the suppressive architecture of the tumor microenvironment, limited trafficking and persistence of therapeutic cells, and the potential for off-tumor toxicity. Furthermore, patient-derived autologous therapies’ logistical and manufacturing complexities constrain their scalability and timely clinical deployment. In contrast, allogeneic strategies offer the potential for standardized, “off-the-shelf” products with broader applicability but require sophisticated genetic engineering to mitigate immunogenicity and preserve therapeutic efficacy.

Advancing the clinical utility of cellular immunotherapy for solid tumors will depend on the continued integration of synthetic biology, gene-editing technologies, and rationally designed combinatorial regimens. These innovations, coupled with rigorous preclinical modeling and robust clinical trial design, will be essential to overcoming current limitations and establishing the next generation of effective, scalable, and safe immunotherapies for solid malignancies.

## Figures and Tables

**Figure 1 ijms-26-05524-f001:**
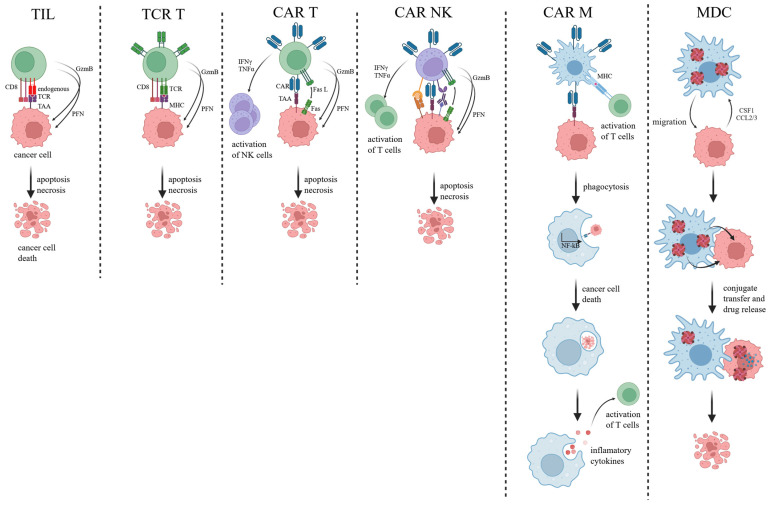
Mechanisms of cell-based therapies against solid tumors described in the review. Tumor-infiltrating lymphocytes (TIL), T cell receptor-engineered T cells (TCR T), chimeric antigen receptor T cells (CAR T), CAR natural killer cells (CAR NK), CAR macrophages (CAR M), and macrophage–drug conjugate (MDC). TILs are naturally tumor-reactive T cells extracted and expanded from a patient’s own tumor, offering polyclonal responses but requiring surgical access. TCR T cells are engineered T cells expressing defined tumor-specific TCRs, enabling precise targeting of intracellular neoantigens but requiring matched HLA and careful validation to avoid off-target toxicity. CAR T cells eliminate cancer cells by recognizing tumor-associated surface antigens (TAA) through their engineered chimeric antigen receptors (CAR), triggering activation, cytokine release, and targeted cytotoxic killing via perforin (PFN) and granzyme (GzmB) pathways. CAR NK cells recognize tumor-specific surface antigens through chimeric antigen receptors, initiating rapid cytotoxic responses via perforin- and granzyme-mediated killing, along with antibody-dependent cellular cytotoxicity (ADCC) and cytokine secretion, contributing to direct tumor clearance and immunomodulation. CAR Ms recognize tumor-associated surface antigens via engineered chimeric antigen receptors, leading to their activation, the phagocytosis of cancer cells, antigen presentation, and pro-inflammatory cytokine release that reshapes the TME and promotes anti-tumor immunity. Macrophage–drug conjugates target and bind to tumor-associated antigens, enabling the macrophage to deliver cytotoxic payloads directly to cancer cells while simultaneously modulating the tumor microenvironment through innate immune activation.

**Figure 2 ijms-26-05524-f002:**
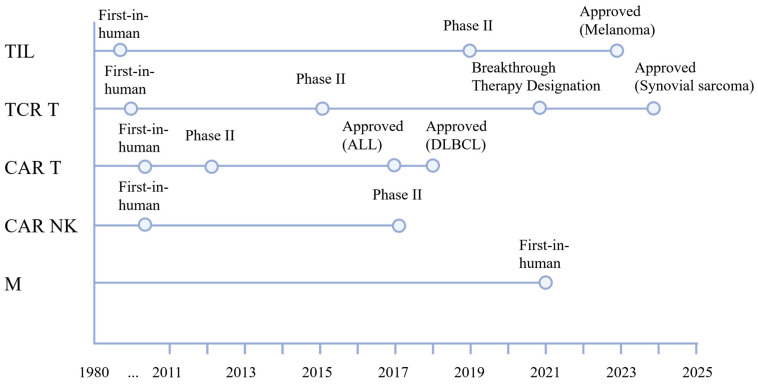
The clinical development milestones for cell-based therapeutic modalities. Tumor-infiltrating lymphocytes (TIL) [27], T cell receptor-engineered T cells (TCR T) [28], chimeric antigen receptor T cells (CAR T) [29,30], CAR natural killer cells (CAR NK) [31], modified macrophages (M). ALL—acute lymphoblastic leukemia, DLBCL—diffuse large B-cell lymphoma.

**Table 1 ijms-26-05524-t001:** Clinical trials of cell-based therapies in solid tumors (clinicaltrials.gov).

Cell Type	Trial Name/ID	ID	Cancer Type(s)	Phase
TIL	NCT05727904	-	Advanced Melanoma	Phase 3
	NCT05361174	-	Melanoma, NSCLC	Phase 1/2
	NCT06481592	-	Endometrial Cancer	Phase 2
	NCT06060613	-	Melanoma, NSCLC, Lung Cancer	Phase 1/2
	NCT05470283	-	Melanoma	Phase 1
TCR T	NCT04044768	MAGE-A4	Synovial Sarcoma	Phase 2
	NCT04526509	NY-ESO-1/LAGE-1a	Various Solid Tumors	Phase 1
	NCT04729543	MAGE-C2	Melanoma, HNSCC	Phase 1/2
	NCT03912831	HPV16 E7	HPV-Associated Cancers	Phase 1
	NCT02650986	NY-ESO-1	Melanoma, Synovial Sarcoma, Ovarian Carcinoma, Peritoneal Carcinoma	Phase 1/2
	NCT00670748	NY-ESO-1	Various Solid Tumors	Phase 2
CAR T	NCT00910650	MART-1	Melanoma	Phase 2
	NCT04581473	Claudin18.2	Gastric, Pancreatic Cancers	Phase 1/2
	NCT02208362	IL13Rα2	Glioblastoma	Phase 1
	NCT04196413	GD2	Diffuse Midline Gliomas	Phase 1
	NCT02706392	ROR1	Triple-Negative Breast Cancer, NSCLC	Phase 1
	NCT01869166	EGFR	Advanced EGFR-positive Solid Tumors	Phase 1/2
	NCT02349724	CEA	Colorectal, Lung, Gastric, Breast, Pancreatic Cancers	Phase 1
	NCT02159716	Mesothelin	Pancreatic, Ovarian, Mesothelioma	Phase 1
	NCT05239143	MUC1	Breast, Ovarian, Pancreatic, Colorectal, Gastric Cancers, NSCLC, HNSCC	Phase 1
	NCT02541370	CD133	Liver, Pancreatic, Brain, Breast, Ovarian, Colorectal Cancers, Hepatocellular Carcinoma	Phase 1/2
	NCT04897321	B7-H3	Pediatric Solid Tumors	Phase 1
CAR NK	NCT06066424	TROP2	Advanced Solid Tumors	Phase 1
	NCT05410717	Claudin6, GPC3, Mesothelin, AXL	Ovarian, Endometrial, Urologic Cancers	Phase 1
	NCT06572956	Various (CAR-T/CAR-NK)	Pancreatic, Prostate, Breast, Glioma	Early Phase 1
CAR M	NCT04660929	HER2	HER2-overexpressing Solid Tumors	Phase 1

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
