# Peer review of "Cell-Based Therapies for Solid Tumors: Challenges and Advances"

_ijms, 2025, doi:10.3390/ijms26125524_

Round 1

Reviewer 1 Report

Comments and Suggestions for Authors

The article titled Cell-based therapies for solid tumors: Challenges and advances” provides an overview of cell-based immunotherapies for solid tumors, effectively contextualizing their mechanisms, clinical applications, and persistent challenges. The authors demonstrate strong scholarly rigor through its integration of preclinical and clinical trial data across multiple modalities (TILs, CAR T/NK cells, macrophages) while maintaining readability for interdisciplinary readers. The review excels in comparing autologous versus allogeneic approaches, providing nuanced insights into manufacturing constraints (e.g., 22.96% CAGR projections vs. individualized processing bottlenecks). However, its analysis of antigen escape mechanisms remains superficial, failing to address emerging solutions like logic-gated CAR systems or synthetic notch receptors. While the TME discussion appropriately highlights stromal barriers, it underrepresents recent advances in ECM-modifying enzymes (e.g., heparanase-engineered CAR T cells). The macrophage section commendably details CAR M mechanisms, but overstates clinical readiness given Carisma Therapeutics' pipeline changes and limited Phase I data [44% stable disease] [NCT04660929]. Economic analyses are inconsistently integrated; while mentioning $1.8 billion oncology market shares, it neglects cost-comparisons between autologous/allogeneic platforms. The conclusion appropriately emphasizes synthetic biology convergence but misses opportunities to discuss spatial transcriptomics for target discovery. Flow disruptions occur in sections like 2.2-2.3 where TCR/CAR overlap isn't clearly delineated, potentially confusing readers about distinct engineering strategies. Although preclinical models are thoroughly cited, there's insufficient critique of their translational limitations in recapitulating human TME complexity. The review makes a strategic error in Section 4 by focusing narrowly on PD-1/PD-L1 while omitting newer checkpoint targets like LAG-3 or TIM-3. While the introduction's cancer statistics are impactful, they lack critical appraisal of histotype-specific response variations to cell therapies. The article's greatest strength lies in its comparative analysis of cellular platforms, particularly the immunological advantages of CAR NK cells versus macrophage trafficking capabilities, though this could be enhanced with head-to-head preclinical data comparisons. The reviewer has the following comments that need to be addressed by authors:

  1. Consider adding a schematic that compares CAR architectures across different immune cell types, such as T cells, NK cells, and macrophages. Annotating key signaling domains in each construct would provide readers with a clear visual summary of structural variations and help contextualize the functional implications of these engineered cells in immunotherapy. This addition would greatly enhance the clarity and educational value of the manuscript.

  1. Including a timeline figure that outlines the clinical development milestones for each therapeutic modality would offer valuable chronological context and help track progress across platforms. Additionally, reorganizing Sections 2.1 to 2.5 using consistent subheadings such as Mechanism, Clinical Data, and Limitations would improve structural coherence and facilitate easier comparison between different approaches.

  1. Expanding the discussion on allogeneic NK cell expansion systems, particularly highlighting advancements in feeder cell technologies, would provide a more comprehensive overview of current manufacturing strategies. This addition would enrich the section by addressing a critical aspect of scalable and clinically viable NK cell therapy development.

  1. Addressing the contradictory data regarding IL15 coexpression in CAR platforms would enhance the scientific rigor of the review. A balanced discussion on the context dependent effects of IL15, such as its role in improving persistence versus potential for cytokine related toxicities, would offer readers a nuanced understanding of its therapeutic implications.

  1. The authors are encouraged to incorporate recent studies demonstrating strategies that enable simultaneous targeting of both tumor cells and the surrounding stromal components, such as cancer-associated fibroblasts. These approaches, along with innovations in adapter-mediated universal CAR platforms, offer promising solutions to overcome the heterogeneity and immunosuppressive barriers characteristic of solid tumors. Including such perspectives would strengthen the discussion on current advances and therapeutic versatility in cell-based treatments for solid malignancies.

https://www.frontiersin.org/journals/immunology/articles/10.3389/fimmu.2025.1539265/full

https://patents.google.com/patent/WO2024191887A2/en?oq=WO2024191887A2

  1. Incorporating 2024 clinical updates on SIRPant M and MDC platforms would enhance the currency and depth of the manuscript. Additionally, adding a focused paragraph on bispecific CAR T cells in Section 2.3, particularly those targeting GD2 and PSMA, would provide a more comprehensive overview of emerging therapeutic strategies and complement the discussed clinical trials.

  1. It would be valuable to define the search strategy and database filters employed for clinical trial inclusion in Table 1 to ensure transparency and reproducibility. Clarifying the rationale for excluding γδ T cells and other emerging effector cell types would further enhance the scope and inclusivity of the review. Additionally, specifying the criteria used to differentiate between descriptors such as "promising" and "limited success" in evaluating efficacy would provide greater clarity and objectivity for readers.

Author Response

Dear Reviewer 1,

We sincerely thank you for the thoughtful comments and valuable suggestions, which have helped us improve the quality and clarity of our manuscript. We have carefully addressed all the points raised and made the corresponding revisions. We hope that the revised version meets the expectations and that the paper is now significantly improved.

Comment1

Consider adding a schematic that compares CAR architectures across different immune cell types, such as T cells, NK cells, and macrophages. Annotating key signaling domains in each construct would provide readers with a clear visual summary of structural variations and help contextualize the functional implications of these engineered cells in immunotherapy. This addition would greatly enhance the clarity and educational value of the manuscript.

Answer1

Thank you for this suggestion. Taking into consideration the suggestions from all of the Reviewers, we prepared a new figure that consists of the mechanisms of action of cell-based therapies described in the review. The figure is placed in the introduction.

Comment2

Including a timeline figure that outlines the clinical development milestones for each therapeutic modality would offer valuable chronological context and help track progress across platforms. 

Answer2

Thank you for this suggestion. We prepared the timeline figure and it is placed in section 2.

Comment3

Additionally, reorganizing Sections 2.1 to 2.5 using consistent subheadings such as Mechanism, Clinical Data, and Limitations would improve structural coherence and facilitate easier comparison between different approaches.

Answer3

Thank you for this comment, the subheadings were added for the easier comparison between the different approaches.

Comment4

Expanding the discussion on allogeneic NK cell expansion systems, particularly highlighting advancements in feeder cell technologies, would provide a more comprehensive overview of current manufacturing strategies. This addition would enrich the section by addressing a critical aspect of scalable and clinically viable NK cell therapy development.

Answer4

Thank you for this suggestion. The discussion on allogenic cells was expanded and the allogenic NK cells are now better described.

Comment5

Addressing the contradictory data regarding IL15 coexpression in CAR platforms would enhance the scientific rigor of the review. A balanced discussion on the context dependent effects of IL15, such as its role in improving persistence versus potential for cytokine related toxicities, would offer readers a nuanced understanding of its therapeutic implications.

Answer5

Thank you for this comment. The discussion on the effects of IL15 was incorporated to the “Advances and challenges” part. 

Comment6

The authors are encouraged to incorporate recent studies demonstrating strategies that enable simultaneous targeting of both tumor cells and the surrounding stromal components, such as cancer-associated fibroblasts. These approaches, along with innovations in adapter-mediated universal CAR platforms, offer promising solutions to overcome the heterogeneity and immunosuppressive barriers characteristic of solid tumors. Including such perspectives would strengthen the discussion on current advances and therapeutic versatility in cell-based treatments for solid malignancies.

https://www.frontiersin.org/journals/immunology/articles/10.3389/fimmu.2025.1539265/full

https://patents.google.com/patent/WO2024191887A2/en?oq=WO2024191887A2

Answer6

Thank you for the suggested studies, they were incorporated into the CAR T cells part.

Comment7

Incorporating 2024 clinical updates on SIRPant M and MDC platforms would enhance the currency and depth of the manuscript. Additionally, adding a focused paragraph on bispecific CAR T cells in Section 2.3, particularly those targeting GD2 and PSMA, would provide a more comprehensive overview of emerging therapeutic strategies and complement the discussed clinical trials.

Answer7

Thank you for this suggestion.

We are very sorry but we couldn’t find any clinical updates on SIRPant – the trial on NHL is ongoing without any results. The same for MDC – the clinical trials are not available. We added a few sentences to those paragraphs.

The paragraph on bispecific CAR T cells was incorporated in Section 2.3.

Comment7

It would be valuable to define the search strategy and database filters employed for clinical trial inclusion in Table 1 to ensure transparency and reproducibility. Clarifying the rationale for excluding γδ T cells and other emerging effector cell types would further enhance the scope and inclusivity of the review. 

Answer7

Thank you, the search was repeated, and the trials have been grouped by the type of cells used as a therapy.

We sincerely hope that all concerns have been adequately addressed in accordance with your suggestions and that the revised manuscript is now clearer and more scientifically robust. We are grateful for your constructive input, which has greatly contributed to improving our work.

Kind Regards

Anna Smolarska, MSc

Reviewer 2 Report

Comments and Suggestions for Authors

This is a well-structured and timely review that provides a critical overview of the current landscape of cell-based immunotherapies for solid tumors. The authors synthesize a vast body of literature covering a range of cellular modalities, including tumor-infiltrating lymphocytes (TILs), TCR-engineered T cells, CAR T cells, CAR NK cells, and genetically modified macrophages. They particularly detailed discussion on the tumor microenvironment, immune evasion mechanisms, and the comparative evaluation of autologous versus allogeneic strategies.

The manuscript is scientifically sound, comprehensive, and well-referenced, offering significant value to the field by summarizing emerging trends and clinical progress while addressing the translational challenges that still limit widespread clinical application. It serves as a valuable resource for researchers, clinicians, and biotech developers working at the intersection of oncology and immunotherapy.

I would only suggest the inclusion of illustrative figures in Section 2 to visually depict the mechanisms of action for each specific cell therapy modality. This would enhance reader comprehension and strengthen the educational value of the review.

Author Response

Dear Reviewer 2,

We sincerely thank you for your comment and valuable suggestion, which have helped us improve the quality and clarity of our manuscript. We hope that the revised version meets the expectations and that the paper is now significantly improved.

Comment1

I would only suggest the inclusion of illustrative figures in Section 2 to visually depict the mechanisms of action for each specific cell therapy modality. This would enhance reader comprehension and strengthen the educational value of the review.

Answer1

Thank you for this suggestion. Taking into consideration the suggestions from all of the Reviewers, we prepared a new figure that consists of the mechanisms of action of cell-based therapies described in the review. The figure is placed in the introduction.

We sincerely hope that all concerns have been adequately addressed in this figure in accordance with your suggestions and that the revised manuscript is now clearer and more scientifically robust. We are grateful for your constructive input, which has greatly contributed to improving our work.

Kind Regards

Anna Smolarska, MSc

Reviewer 3 Report

Comments and Suggestions for Authors

This is a well-written paper that summarises the literature of the cell-based therapies for solid tumors, and explores the obstacles and advances fro implementing these approaches. I have a number of comments, albeit they are all minor issues.

  1. Figure 1 - this figure seems to add too little information to the paper. I suggest modifying to add more detail on how each cell-therapy is unique and what are the key characteristics of each approach in inducing cytotoxic activity.
  2. Li 102 - be more specific about which TME barrier you are referring to.
  3. Passage in Li 120-125 should be moved to the beginning, after the first passage in section 2.1.
  4. Li 137 -give more specific samples on the innovations.
  5. Li 141-143 -this text seems redundant.
  6. Section 2.2. title - add abbreviation in the brackets (TCR T). 
  7. The first paragraph in the section 2.2. should be rephrased to better explain the types of cells that are the topic of this section. It's not entirely clear which parts refer to TCR T and which on TRC.
  8. Section 2.3.: title - add full expression before the abbreviation; li 192-196 speak of T lymphocytes in general and can be moved after the section 2 title.
  9. Li 200 - add a few lines on the specific antigens that are targeted here, and also in the paragraphs between lines 229-235. 
  10. Are there any studies on successful uses of CAR t cell-based therapy in solid cancers?
  11. Li 249-251 - can be removed as they seem redundant.
  12. Li 275-280 -transfer to the beginning of Section 2.4.
  13. Li 440 How successful are these mitigation measures? Expand a little bit if possible.

Author Response

Dear Reviewer 3,

We sincerely thank you for the thoughtful comments and valuable suggestions, which have helped us improve the quality and clarity of our manuscript. We have carefully addressed all the points raised and made the corresponding revisions. We hope that the revised version meets the expectations and that the paper is now significantly improved.

Comment1

Figure 1 - this figure seems to add too little information to the paper. I suggest modifying to add more detail on how each cell-therapy is unique and what are the key characteristics of each approach in inducing cytotoxic activity.

Answer1

Thank you for this suggestion. Taking into consideration the suggestions from all of the Reviewers, we prepared a new figure that consists of the mechanisms of action of cell-based therapies described in the review. The figure is placed in the introduction.

Comment2

Li 102 - be more specific about which TME barrier you are referring to.

Answer2

Thank you for your comment. The TME barriers are now better described in sections 2.1 and 2.1.2.

Comment3

Passage in Li 120-125 should be moved to the beginning, after the first passage in section 2.1.

Answer3

Thank you, this pert has been moved to the beginning of the Mechanism part (section 2.1.1)

Comment4

Li 137 -give more specific samples on the innovations.

Answer4

Thank you, the innovations have been described in the 2.1.2 section.

Comment5

Li 141-143 -this text seems redundant.

Answer5

Thank you, it has been removed

Comment6

The first paragraph in the section 2.2. should be rephrased to better explain the types of cells that are the topic of this section. It's not entirely clear which parts refer to TCR T and which on TRC.

Answer6

Thank you for this suggestion. The paragraph has been changed to be more understandable.

Comments7

Section 2.2. title - add abbreviation in the brackets (TCR T).

Section 2.3.: title - add full expression before the abbreviation;

li 192-196 speak of T lymphocytes in general and can be moved after the section 2 title.

Answer7

Thank you, we have addressed all the comments and suggestions provided by the reviewer and revised the manuscript accordingly.

Comment8

Li 200 - add a few lines on the specific antigens that are targeted here, and also in the paragraphs between lines 229-235.

Answer8

Thank you, the specific antigens have been listed in paragraphs 2.3.1 and 2.3.3.

Comment9

Are there any studies on successful uses of CAR t cell-based therapy in solid cancers?

Answer9

Yes, thank you, this information was incorporated in paragraph 2.3.2

Comments10

Li 249-251 - can be removed as they seem redundant.

Li 275-280 -transfer to the beginning of Section 2.4.

Answer10

Thank you, we have addressed all the comments and suggestions provided by the reviewer and revised the manuscript accordingly.

Comment11

Li 440 How successful are these mitigation measures? Expand a little bit if possible.

Answer11

Thank you for this comment. Details have been added to section 5.

We sincerely hope that all concerns have been adequately addressed in accordance with your suggestions and that the revised manuscript is now clearer and more scientifically robust. We are grateful for your constructive input, which has greatly contributed to improving our work.

Kind Regards

Anna Smolarska, MSc